# The electrical heart axis of the fetus between 18 and 24 weeks of gestation: A cohort study

**Carlijn Lempersz**[1,2]*, **Lore Noben**[1,2], **Sally-Ann B. Clur**[3], **Edwin van den Heuvel**[4], **Zhouzhao Zhan**[4], **Monique Haak**[5], **S. Guid Oei**[1,2,6], **Rik Vullings**[6], **Judith O. E. H. van Laar**[1,2,6]

1 Department of Obstetrics and Gynecology, Máxima Medical Center, Veldhoven, The Netherlands,
2 Eindhoven MedTech Innovation Center (e/MTIC), Eindhoven, The Netherlands, 3 Department of Pediatric Cardiology, Amsterdam University Medical Center, Amsterdam, The Netherlands, 4 Department of Mathematics and Computer Science, Eindhoven University of Technology, Eindhoven, The Netherlands, 5 Department of Obstetrics and Gynecology, Leiden University Medical Center, Leiden, The Netherlands, 6 Department of Electrical Engineering, Eindhoven University of Technology, Eindhoven, The Netherlands

* c.lempersz@gmail.com

**Data Availability Statement:** Data cannot be shared publicly because this was not included in the informed consent procedure participants signed for. Data are available from the Máxima

## Abstract

## Introduction

A fetal anomaly scan in mid-pregnancy is performed, to check for the presence of congenital anomalies, including congenital heart disease (CHD). Unfortunately, 40% of CHD is still missed. The combined use of ultrasound and electrocardiography might boost detection rates. The electrical heart axis is one of the characteristics which can be deduced from an electrocardiogram (ECG). The aim of this study was to determine reference values for the electrical heart axis in healthy fetuses around 20 weeks of gestation.

## Material and methods

Non-invasive fetal electrocardiography was performed subsequent to the fetal anomaly scan in pregnant women carrying a healthy singleton fetus between 18 and 24 weeks of gestation. Eight adhesive electrodes were applied on the maternal abdomen including one ground and one reference electrode, yielding six channels of bipolar electrophysiological measurements. After removal of interferences, a fetal vectorcardiogram was calculated and then corrected for fetal orientation. The orientation of the electrical heart axis was determined from this normalized fetal vectorcardiogram. Descriptive statistics were used on normalized cartesian coordinates to determine the average electrical heart axis in the frontal plane. Furthermore, 90% prediction intervals (PI) for abnormality were calculated.

## Results

Of the 328 fetal ECGs performed, 281 were included in the analysis. The average electrical heart axis in the frontal plane was determined at 122.7˚ (90% PI: -25.6˚; 270.9˚).

Medical Center Institutional Data Access (contact via jolanda.luime@mmc.nl) for researchers who meet the criteria for access to confidential data.

**Funding:** This research was supported by The Dutch Technology Foundation STW (#12470), Stichting de Weijerhorst, Horizon2020 (#719500). The funders had no role in study design, data collection and analysis, decision to publish, or preparation of the manuscript.

**Competing interests:** R. Vullings is a shareholder in Nemo Healthcare BV, the Netherlands. S.G. Oei initiated the scientific research from which Nemo Healthcare originated, there is no financial relationship between Nemo Healthcare and S.G. Oei. All other authors have declared that no competing interests exist.

## Discussion

The average electrical heart axis of healthy fetuses around mid-gestation is oriented to the right, which is, due to the unique fetal circulation, in line with muscle distribution in the fetal heart.

## Introduction

In developed countries a fetal anomaly scan in mid-pregnancy is performed to check for the presence of congenital anomalies, including congenital heart disease (CHD). The importance of prenatal CHD detection has been highlighted by was previous research that found a reduction in neonatal morbidity and mortality when CHD was diagnosed prenatally [1, 2]. The introduction of a standardized screening program for the fetal anomaly scan in mid-pregnancy has led to an increase in prenatal CHD detection rates in the Netherlands up to 40–60%. However, 40% of CHD is still missed [3]. Ultrasound detection of CHD is difficult due to fetal body movements, and the small size and rhythmic movements of the fetal heart. Furthermore, detection rates depend on the experience of the sonographer, fetal position and BMI of the mother [4–13]. New diagnostic tools are needed to further increase the prenatal detection of CHD.

A tool might be the non-invasive fetal electrocardiogram (NI-fECG). NI-fECG enables the production of a 12-lead electrocardiogram by means of a standardized method [14]. ST-segment elevations are seen in ischemia and deviation of the electrical heart axis occurs in some cardiac malformations (e.g. hypoplastic right heart syndrome, atrioventricular septal defect) [15–17]. The electrical heart axis is one of the characteristics which can be deduced from an ECG. It represents the median vector of the electrical activity through the heart during one cardiac cycle and provides information about the muscle distribution of the heart.

Verdurmen et al. found a right-oriented electrical heart axis in healthy fetuses [18]. This has also been described in term fetuses during labor and in neonates [19, 20]. The right-oriented electrical heart axis in healthy fetuses can be explained by the fetal circulation that has a unique physiology with multiple shunts to bypass the lungs, so that the right ventricle pumps 60% of the cardiac output, leading to a right ventricular dominance. After birth the pulmonary vascular resistance drops and the venous return to the left atrium increases leading to an increase in the cardiac output of the left ventricle. The left ventricle pumps against the high resistance systemic system once the placental circulation is eliminated [21]. With time the left ventricular muscle mass gradually increases and a leftwards shift of the electrical heart axis occurs. We hypothesize that the presence of certain CHD can already cause a deviated electrical heart axis in utero.

The aim of this paper was to determine reference values for the electrical heart axis in mid-term healthy fetuses.

## Materials and methods

The study protocol was previously published by Verdurmen et al. [22] Ethical approval by the institutional review board of the Máxima Medical Center was obtained before enrolment (NL48535.015.14). Fetal ECG measurements were performed from May 2014 until September 2018 at the Máxima Medical Centre Veldhoven, The Netherlands, a tertiary care referral center for obstetrics and at 'Diagnostiek voor U' diagnostic center, Eindhoven, The Netherlands.

## Study population

Pregnant women carrying a singleton fetus without known congenital anomalies and a gestational age between 18 and 24 weeks of gestation were included. All patients were older than 18 years and gave written informed consent prior to the fetal ECG measurement.

Patients who did not understand the Dutch language well and/or had multiple pregnancies were excluded. If CHD was found later in pregnancy or after birth, the measurement was excluded from analysis.

The following data was gained prospectively: maternal gravidity and parity, as well as obstetric and general medical history. Parents received a questionnaire three months after birth to confirm that the child was healthy and did not have any congenital diseases. We chose this three-month cut-off point as at this age, all children in the Netherlands have had their second medical check-up by a doctor, who, among other things, evaluates cardiac health using auscultation.

## fECG measurements and signal processing

Singular fetal ECG measurements were performed subsequent to the fetal anomaly scan. Women lay in a semi-recumbent position to prevent aortocaval compression. To yield six channels of bipolar electrophysiological measurements, eight electrodes were placed on the maternal abdomen in a fixed configuration. Two electrodes served as common reference and ground electrodes respectively (Fig 1; [23]). Before application of the electrodes the skin was washed with water and soap after which skin preparation was performed with medical abrasive paper (Red DotTM Trace Prep, 3M Health Care, Ontario, Canada) to optimize skin

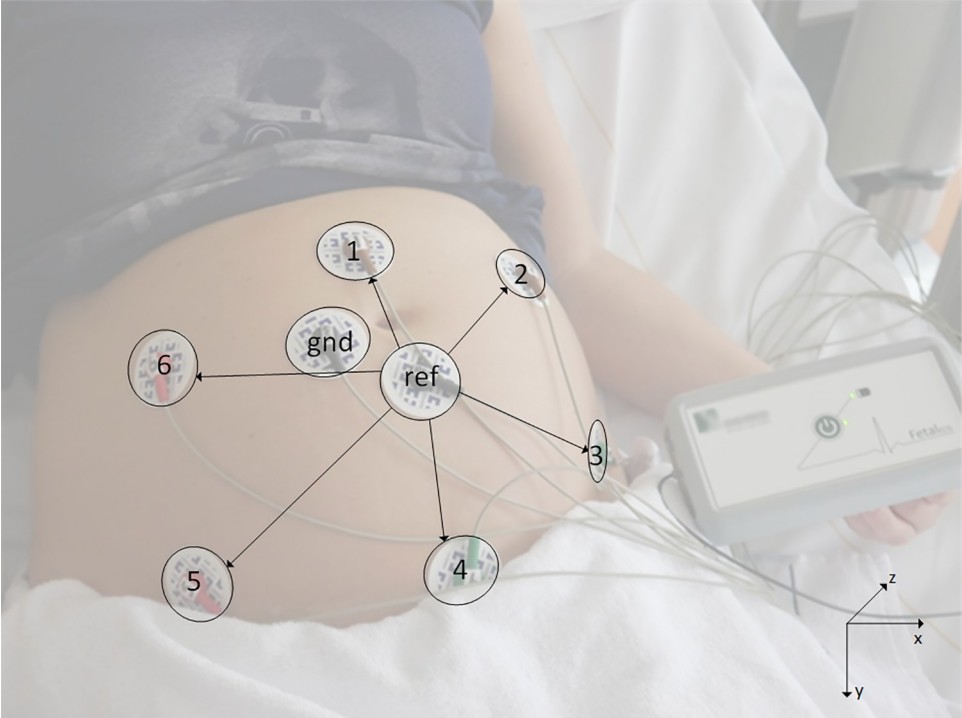

**Fig 1. Measurement set-up of the non-invasive fetal electrocardiogram.** Eight electrodes were placed on the maternal abdomen in a fixed configuration. Two electrodes served as common reference (Ref) and ground (Gnd). The cartesian coordinate system as used in our analyses is displayed in the bottom right corner [23].

impedance. Each measurement lasted around 30 minutes during which fetal orientation was ultrasonographically checked at four fixed time intervals. After training by an experienced gynecologist or sonographer the researcher determined the fetal orientation by ultrasound. The fetal orientation was determined following a step-by-step plan in which the spine was taken as an identifiable landmark. The ultrasound probe was held only in a horizontal and/or vertical position for it to be reproducible and annotations were made about the position of the probe. To evaluate the accuracy of our orientation correction, the correct ECGs/VCGs of a single participant are compared to verify their consistency.

Fetal ECG measurements were performed with a 6-channel electrophysiological amplifier (Nemo Healthcare BV, The Netherlands) using adhesive Ag/AgCl electrodes (Red DotTM, 3M Health Care, Ontario, Canada) on the maternal abdomen. The measured electrophysiological signals were digitized at 500 Hz sampling frequency and stored on a computer for offline analysis.

This offline analysis consisted of a series of signal processing steps, aimed to suppress interferences and standardize the fetal ECG signals for fetal orientation, so that the fetal electrical heart axis could be measured. These signal processing steps have been described in more detail in Lempersz et al. 2020 [23]. In the first step of signal processing, interferences from the maternal ECG, abdominal muscles, and extracorporal sources were suppressed by an adaptive template-based method [24]. As a result, for each of the six recorded signals a fetal ECG signal was obtained, yet at relatively low signal-to-noise ratio. Because each fetus could have a different orientation with respect to the maternal abdomen and the recording electrodes placed on this abdomen, the fetal ECG signals not only changed between participants, but also within participants due to fetal movement.

The second step in the signal processing aimed to standardize for fetal orientation. To allow for such standardization, a fetal vectorcardiogram was calculated for every heartbeat first, combining the information from the six abdominal signals into a 3-dimensional fetal ECG complex [25]. This vectorcardiogram could subsequently be tracked over time, detecting fetal movements and correcting for them by rotating the fetal vectorcardiogram in 3-dimensional space. Finally, another rotation in 3-dimensional space was applied that corrected for the fetal orientation, which was assessed from intermittent ultrasound scans. For instance, if the ultrasound indicated that the fetus was in a cephalic position, the recorded fetal vectorcardiogram was rotated by 180 degrees to represent the fetal vectorcardiogram as if the fetus was in a breech position, mimicking the position used when making adult ECGs. Similarly, the fetus was rotated along the longitudinal axis as if the fetal back was orientated towards the back of the mother. The parts of the measurements of sufficient signal quality, closest to the performance of the ultrasound determining fetal orientation, were used to create the vectorcardiogram.

Finally, to enhance the signal-to-noise ratio, orientation-standardized fetal vectorcardiograms were averaged over multiple heartbeats to yield one fetal vectorcardiogram per measurement.

The orientation of the electrical heart axis was defined as the direction in which the vectorcardiogram had its maximum amplitude [25]. The latter direction was estimated as the average direction of the dominant vectors in the QRS complex, defined as the vectors from the point that the R-wave exceeded 70% of its maximum value until the point that it fell below 70% of the maximum value. The orientation of the fetal heart axis was expressed in degrees ranging from minus 180˚ to plus 180˚ and calculated in the frontal plane, where minus 90˚ is located superiorly.

## Statistical analysis

The observed frontal angle was determined in the (x,y)-plane. The normalized coordinates $(\tilde{x}, \tilde{y})$ were calculated as the division of the originate coordinates (x, y) by their Euclidean norm $\sqrt{x^2 + y^2}$.

We calculated descriptive statistics (median with interquartile range (IQR)) on the normalized $(\tilde{x}, \tilde{y})$ Cartesian coordinates. We also reported the average frontal axis with 90% prediction intervals that would function as reference values. Prediction intervals are chosen because they account for the uncertainty in estimating the population mean and the random variation of the individual values [26]. The average frontal axis is a circular mean, therefore the maximum likelihood estimate (mle) of the mean direction parameter $\mu$ of a von Mises distribution will be used as the average frontal axis.

$$\bar{\theta} = \text{atan2}(\bar{S}/\bar{C}), \text{ where } \bar{C} = \frac{1}{n}\sum_{j=1}^{n}\cos\theta_j \text{ and } \bar{S} = \frac{1}{n}\sum_{j=1}^{n}\sin\theta_j$$

The prediction intervals were calculated, using the lower and upper quantiles of the Von Mises distribution with the estimated parameters.

Statistical analysis was conducted with SAS (version 9.4, SAS Institute Inc., NC, USA) and R (version 3.5.3, R Foundation, Vienna, Austria). Descriptive statistics (median with interquartile ranges) were used to describe baseline characteristics, using IBM SPSS statistics version 25.0 (SPSS Inc., Chicago, Ill., USA).

Data are available upon request.

## Results

A total of 328 patients were included. From these, 15 measurements were excluded due to missing or incomplete questionnaires and 23 measurements were excluded due to missing information on the fetal orientation. CHD was found in one neonate and a chromosomal disorder was present in three neonates as reported in the postpartum questionnaire, necessitating their exclusion. Of the remaining 286 inclusions (87.2% of the original 328 included patients), five measurements had to be excluded due to poor quality NI-fECG recordings. A total of 281/286 measurements were available for further analysis giving a success rate of 98%. Table 1 shows the characteristics of the study population. Fig 2 shows a flowchart of the included measurements.

Fig 3 is an example of a fetal electrocardiogram, here one can see a clear QRS-complex.

The median and interquartile range (IQR) of the $\tilde{x}$ coordinate was 0.347 (1.660) and that of the $\tilde{y}$ coordinate was 0.327 (0.956). Based on these normalized coordinates, the average frontal angle was determined at 122.68˚ (90% PI: -25.6˚; 270.9˚). Fig 4 shows the distribution of the orientation of the electrical heart axis of each fetus. The arrow shows the mean electrical heart axis with, in grey, corresponding 90% PI in the frontal plane.

**Table 1. Baseline characteristics of participants (N = 281).**

|  | Mean (± SD) |
|---|---|
| **Age (years)** | 31.3 (± 4.0) |
| **GA (weeks)** | 20.2 (± 1.3) |
| **Nulliparous (%)** | 52.3 |
| **BMI (kg/m$^2$)** | 24.4 (± 5.4) |

Abbreviations: GA = gestational age, BMI = body mass index.

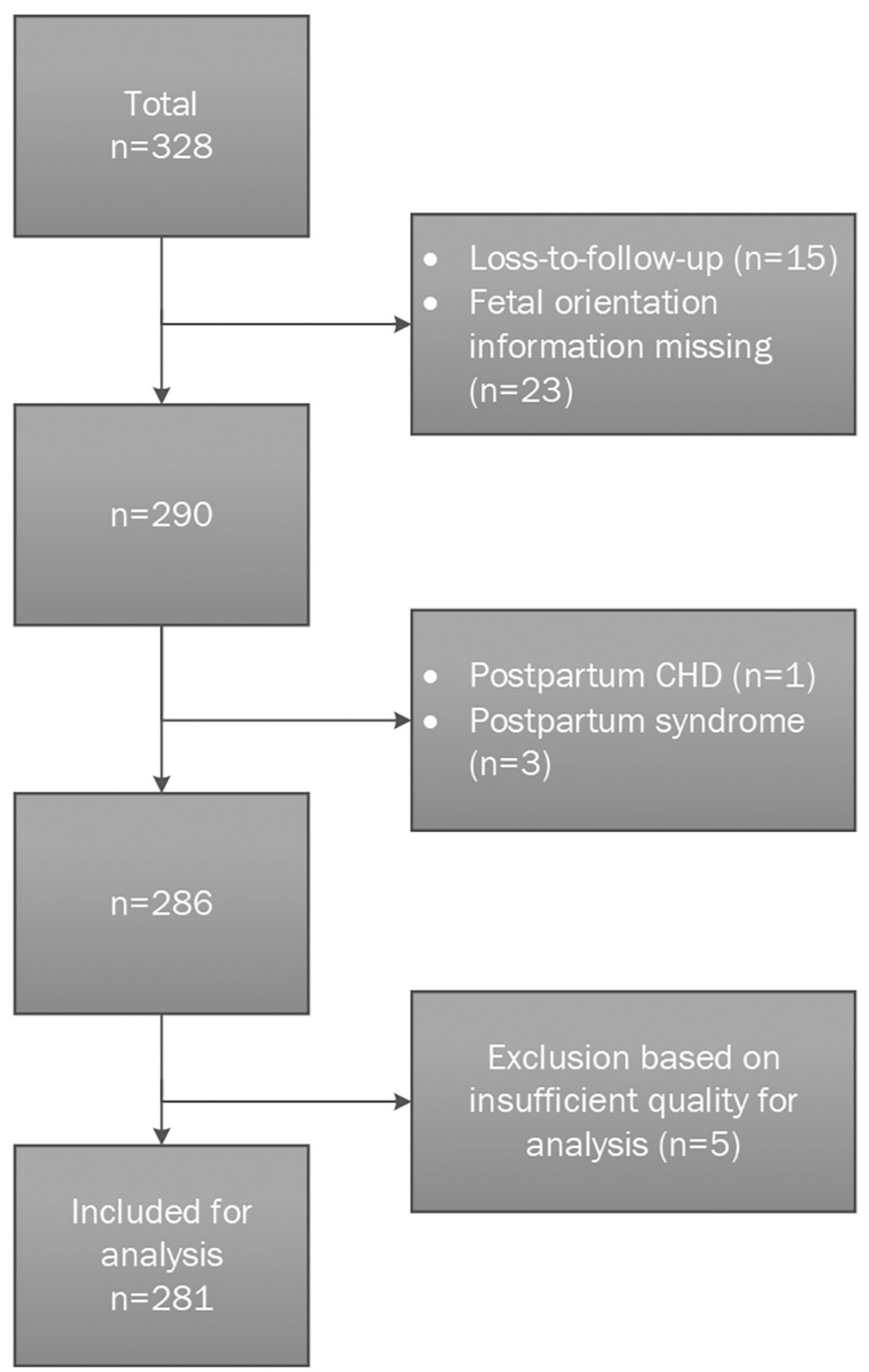

**Fig 2. Flowchart of the included measurements.**

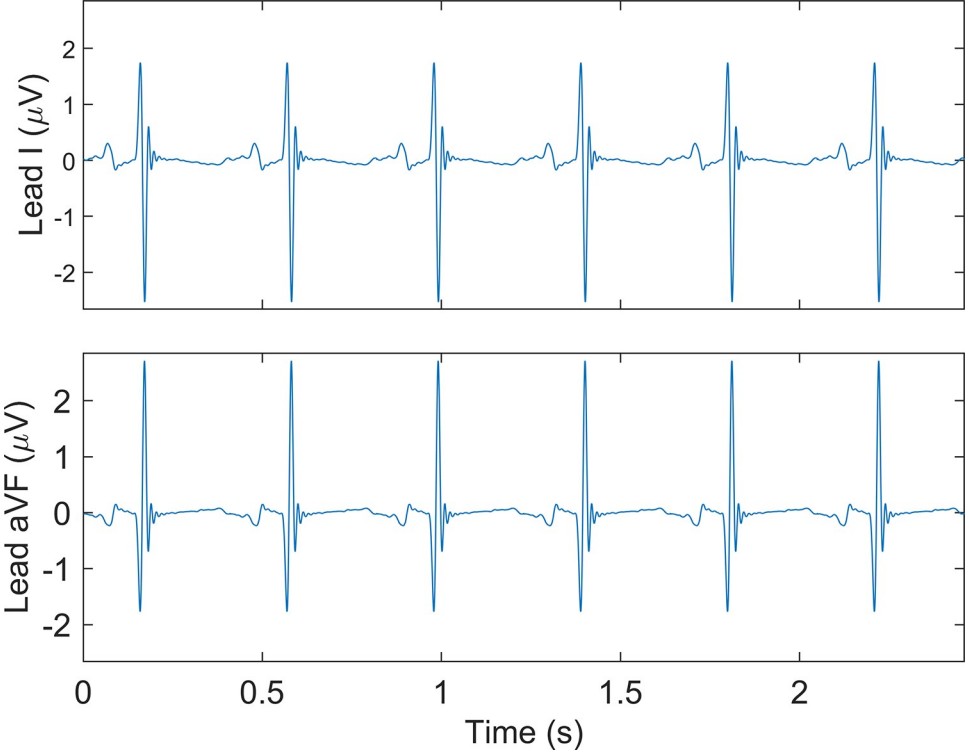

**Fig 3. Example of a fetal electrocardiogram.** Lead I and aVF. x-axis is time in seconds (s), y-axis is electric potential in microvolts (μV).

## Discussion

### Main findings

In this paper we present reference values for the electrical heart axis calculated from our cohort of 281 healthy fetuses at mid-gestation. We found the mean electrical heart axis of the healthy fetus orientated to the right (122.68˚), which is in line with the distribution of fetal cardiac muscle mass due to the unique anatomy of the fetal circulatory system and findings from previous studies [17–20, 27]. We found that the prediction intervals based on our cohort are wide, indicating a broad range wherein future observations will fall.

### Strengths and limitations

The main strength of this study is the large group of participants and the low number of recordings excluded due to inadequate data quality. The latter shows that this technology has improved significantly compared to earlier reported research [28–30]. This high success rate is an indispensable characteristic for any technology to be implemented in daily practice. However, the time needed to process the recordings is at this moment the limiting factor for the NI-fECG technology, which currently still takes place offline. Therefore, results are not yet readily available during the measurement. This can be solved by automatization of the signal processing algorithms in the future which can then be incorporated in the measurement hardware. Furthermore, a few factors can contribute to inaccuracies in the correction for the fetal orientation; inaccuracies in the correction for the fetal movement and inaccurate assessment of the fetal orientation and the time at which this assessment was made. To minimize these inaccuracies, we have used the ECG data that was recorded around the time that the fetal

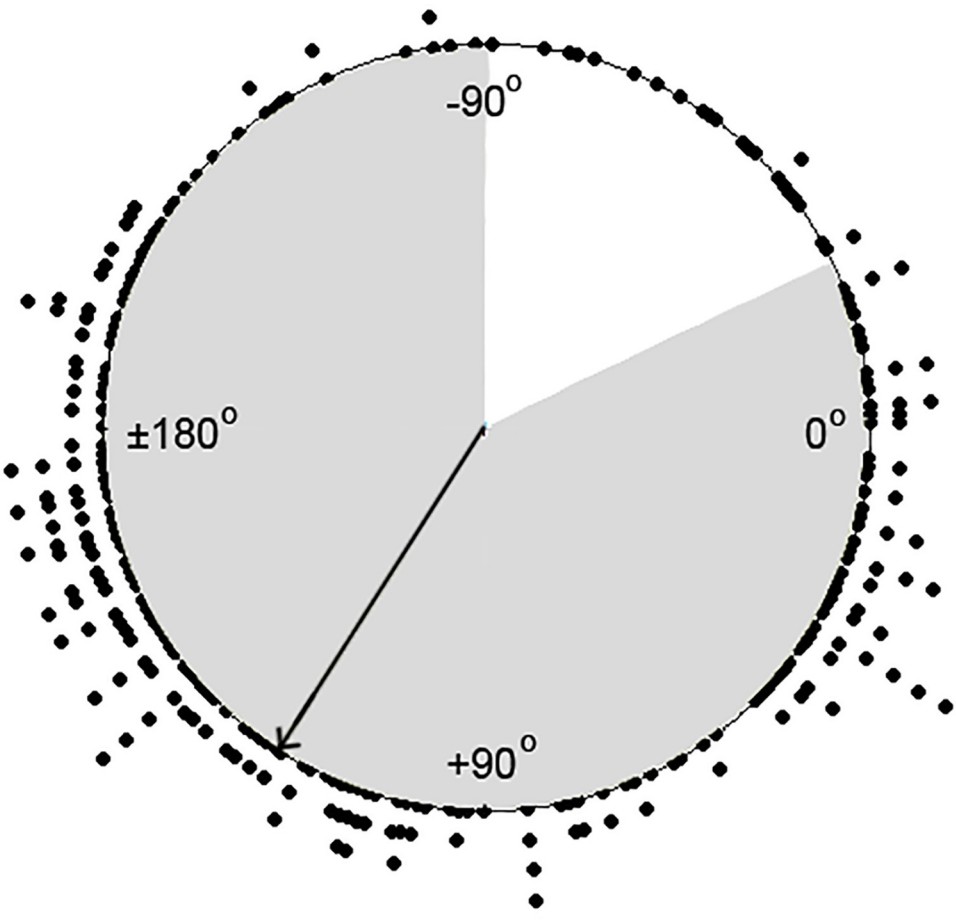

**Fig 4. Distribution of the orientation of the electrical heart axis plotted in a circle diagram.** Each dot represents one fetus. The arrow represents the mean electrical heart axis with corresponding 90% PI in the frontal plane in grey.

orientation assessment was done. Possible cumulative errors in the correction for movement will therefore be limited. As an extra check, the fetal orientation was determined multiple times per recording. Within a single patient, the correct ECGs/VCGs are compared to verify their consistency, which can act as an indirect evaluation of the accuracy of our orientation correction.

## Interpretation

To our knowledge, this is the first study that determines reference values for the electrical heart axis in healthy midterm fetuses. Recent advances in the signal processing algorithms have made it possible to acquire information on the fetal ECG in the antenatal period in a non-invasive manner. This makes it possible to define reference values for the electrical heart axis in healthy fetuses in mid-pregnancy.

The electrical heart axis reflects the distribution of muscle mass in the fetal heart. In the fetal circulation with its three obligatory shunts and the high resistance pulmonary and low resistance systemic circulations, the right ventricle is dominant and pumps about 60% of the cardiac output. As a consequence the muscle mass of the right ventricle is greater than that of

the left ventricle and this results in greater amplitude of depolarization together with decreased speed of depolarization on the right side [31]. Our results confirm this right oriented electrical heart axis in healthy fetuses. The next step towards determining the use of this parameter for screening purposes is to define the electrical heart axis in fetuses with CHD.

Fetal electrocardiography is an easy to use, non-invasive, safe technology with a minimal burden for the pregnant women. Further research towards the electrical heart axis in fetuses with different types of CHD is necessary to determine which defects are associated with a deviated fetal electrical heart axis. Then the NI-fECG could be performed in addition to the fetal anomaly scan around the 20[th] week of gestation as part of prenatal screening after automatization of the signal processing of the recording. A point of attention is the broad distribution of the electrical heart axis found in our cohort of healthy fetuses in mid-pregnancy. This resulted in wide predictions intervals [-25.6˚; 270.9˚] making the use of the electrical heart axis alone as a parameter for the screening of CHD less suitable. Future research towards ECG waveform and ECG intervals may add to the development of additional ECG parameters which could further enhance the prenatal detection of CHD.

## Conclusion

Our results confirm that the mean electrical heart axis of healthy fetuses around mid-gestation is oriented to the right. The wide prediction interval for the frontal heart axis found in our cohort, is unfavorable for future implementation of this method for screening purposes. Further research towards the electrical heart axis in fetuses with CHD as well as additional ECG waveform and intervals may elucidate the role of fetal ECG as a screening parameter for the detection of CHD.

## Supporting information

**S1 Checklist. TREND statement.**
(PDF)

**S1 File. Study protocol.**
(DOCX)

## Acknowledgments

The authors would like to express their gratitude to ´Diagnostiek voor U´ diagnostic center, (Eindhoven, The Netherlands), N. Eijsvoogel, D. Aben, O. Hulsenboom, M. Sengers, J. Drinkwaard, C. van den Oord, M. van Wierst, L. Cornelissen, C. de Vet and M. van Bruggen for their cooperation and effort towards the recruitment of patients and their role in the data collection.

## Author Contributions

**Conceptualization:** Carlijn Lempersz, S. Guid Oei, Rik Vullings, Judith O. E. H. van Laar.

**Data curation:** Carlijn Lempersz, Lore Noben, S. Guid Oei, Rik Vullings, Judith O. E. H. van Laar.

**Formal analysis:** Edwin van den Heuvel, Zhouzhao Zhan, Rik Vullings.

**Funding acquisition:** S. Guid Oei, Rik Vullings.

**Investigation:** Carlijn Lempersz, Lore Noben, Sally-Ann B. Clur, Monique Haak, Judith O. E. H. van Laar.

**Methodology:** Carlijn Lempersz, Edwin van den Heuvel, Zhouzhao Zhan, Judith O. E. H. van Laar.

**Project administration:** Carlijn Lempersz, Lore Noben, Sally-Ann B. Clur, Monique Haak, Judith O. E. H. van Laar.

**Software:** Edwin van den Heuvel, Rik Vullings.

**Supervision:** S. Guid Oei, Judith O. E. H. van Laar.

**Writing – original draft:** Carlijn Lempersz.

**Writing – review & editing:** Lore Noben, Sally-Ann B. Clur, Zhouzhao Zhan, Monique Haak, S. Guid Oei, Rik Vullings, Judith O. E. H. van Laar.

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
