## [Decision Letter · Decision Letter 0]

6 Apr 2021

PONE-D-21-02858

The electrical heart axis of the fetus between 18 and 24 weeks of gestation: a cohort study.

PLOS ONE

Dear Dr. Lempersz,

Thank you for submitting your manuscript to PLOS ONE. After careful consideration, we feel that it has merit but does not fully meet PLOS ONE’s publication criteria as it currently stands. Therefore, we invite you to submit a revised version of the manuscript that addresses the points raised during the review process.

We look forward to receiving your revised manuscript.

Kind regards,

Gabor Erdoes, M.D., Ph.D.

Academic Editor

PLOS ONE

Journal Requirements:

4. We note that you have not provided a financial disclosure with the original submission data.

5. We note that you did not complete the Competing Interests section with the original submission data.

a. Please complete your Competing Interests statement to state any Competing Interests. If you have no competing interests, please state "The authors have declared that no competing interests exist.", as detailed online in our guide for authors at http://journals.plos.org/plosone/s/submit-now

6. Please include captions for your Supporting Information files at the end of your manuscript, and update any in-text citations to match accordingly. Please see our Supporting Information guidelines for more information: http://journals.plos.org/plosone/s/supporting-information

Additional Editor Comments:

Please especially provide detailed answers to the comments of reviewer 3, as this reviewer was particularly critical of the study.  

Reviewers' comments:

Reviewer's Responses to Questions

**Comments to the Author**

1. Is the manuscript technically sound, and do the data support the conclusions?

Reviewer #1: Yes

Reviewer #2: Yes

Reviewer #3: Yes

2. Has the statistical analysis been performed appropriately and rigorously? 

Reviewer #1: Yes

Reviewer #2: Yes

Reviewer #3: Yes

3. Have the authors made all data underlying the findings in their manuscript fully available?

Reviewer #1: No

Reviewer #2: Yes

Reviewer #3: Yes

4. Is the manuscript presented in an intelligible fashion and written in standard English?

Reviewer #1: Yes

Reviewer #2: Yes

Reviewer #3: No

5. Review Comments to the Author

Reviewer #1: This study examined the electrical axis of the fetal heart between 18 and 24 weeks gestation in a cohort study. This was a normative study in fetuses shown to have a normal fetal echo at a mean of 20 weeks gestation. Utilizing a non-invasive method with NEMO technology resulted in 6 dipolar electrodes in a standardized array on the maternal abdomen. Fetal echo orientation of the body position allowed the generation of a fetal vectorcardiogram from which a mean frontal fetal cardiac electric axis was generated equaling 123 degrees with 90% PI of 26 – 271 degrees. They concluded that this is consistent with the expected rightward electrical axis of the normal fetus and that they have generated reference values. While there is indeed a great need to perfect non-invasive fetal ecg’s for research and clinical purposes, and the authors are making a great contribution to science by developing this technology, unfortunately, the very wide confidence intervals of the mean value, even with such a large cohort, implies that this particular parameter is not clinically useful as is, and this is acknowledged by the authors. However, they are encouraged to continue to explore the utility of this technique. In fact, they have included a copy of their large long-term pilot study of fetal non-invasive electrocardiograms, of which this project is a small part.

It was unexpected that the electrical axis turned out to be so variable in this study. I posit that the greatest source of error is in their estimation of the fetal chest position. The mean frontal electric cardiac axis is very dependent on the orientation of the heart in relation to the 3-dimensional thorax. Postnatally, our electrode arrays are oriented to the usual levocardia, can be corrected for dextrocardia, and are known to vary with altered positional anatomy of the heart (eg, mesocardia) and the thorax (eg, scoliosis). The study’s method of having trainees periodically assess the fetal lie during the ecg, while the fetus may be moving, is the probable source of error. It is neither accurate nor precise. I would humbly suggest that they try this again, but next time carefully correct the vectorcardiogram for a fixed, anatomically and easily identified landmark, such as diaphragm, or even spine. Shortening the length of study might also reduce error due to fetal movement.

A few questions and comments:

Was this a convenience sample, or all consecutive normal fetal echos?

The technical success rate was remarkable in that only 5 of 281 eligible fetuses could not be analyzed.

The Title and Abstract are good.

The Introduction is long, but clear and informative.

The Methods are adequate, and while they mostly refer to their previous publications, these and the Research Protocol are all readily available.

The Results are straightforward.

The Discussion is a bit lengthy, but complete, including strengths, weaknesses, and future directions.

The text is clear and well written, but still needs careful proofreading.

This group has many prior publications in this field, which is supportive.

The References are comprehensive and relevant.

The Figures and Tables are good.

Overall, the science appears to be valid, technically believable and thus reproducible. The results are unique. Again, I encourage the authors to improve their correction for fetal thorax orientation.

Specific Comments:

The copy of the manuscript provided to me somehow included all of the preceding editorial markups and corrections. In addition, the text version I accessed included prior assessments by 3 previous reviewers. However, I generated my review before reading the other opinions.

Reference 30 is a book and appears incomplete without the name and location of the publisher.

The word “data” is always plural.

The sentence on lines 163-165 is unclear to me.

Reviewer #2: Thank you for the opportunity to read this interesting paper about fetal ECG´s performed in normal pregnancies. The authors conclude, that the range of a normal fetal heart axis is wide and the fetal heart axis alone is not suitable for screening of CHD due to the wide prediction interval. I agree to the comments of the three previous reviewers. The comments of the authors and their changes within the text improved the paper. In conclusion, the authors showed that despite an optimal setting (study setting, optimal time window) the fetal heart axis is wide [and its use not applicable for screening of CHD].

Although they did not include fetuses with CHD als reference group, it many be reasonable to draw the above mentioned conclusions.

This should be published.

comment to the revision: lines 292 to 299: The statement that the heart axis has been shown to rotate in a fetus with aortic stenosis or critical pulmonary stenosis is an assumption that has not yet been proven. A fetal echo has a significantly greater significance here. I recommend deleting this section.

Reviewer #3: Lempersz et al performed a prospective study to evaluate the normal cardiac axis in fetuses.

Abstract:

This study does not evaluate electrical heart axis in CHD, it only evaluates normal fetuses. Please do not put in the discussion anything regarding the electrical axis in CHD screening.

Introduction:

Please cite: NI-fECG enables the production of a 12-lead electrocardiogram by means of a standardized method.

Methods:

Study population, how do doctors evaluate cardiac health? Do they do echocardiograms?

I find this paragraph from 124-130 confusing. Either describe the protocol in full or leave it out.

Results/conclusions:

I think the major limitation of this study is the utility of this information. I think using this information to then compare to patients with CHD should be the next step and would make this a much more robust and useful study. Otherwise, the study should be reframed as a "proof of concept" study since we know that neonates have a rightward axis. I don't think any information regarding CHD should be put in the paper, because it misleads the reader into thinking that there will be some information on the fetal electrical axis in CHD.

Further comments:

Previous edits are seen in the PDF, which made it challenging to review.

6. PLOS authors have the option to publish the peer review history of their article (what does this mean?). If published, this will include your full peer review and any attached files.

Reviewer #1: No

Reviewer #2: **Yes: **Ulrike Herberg

Reviewer #3: No

---

## [Author Response · Author response to Decision Letter 0]

20 May 2021

5. Review Comments to the Author

Reviewer #1: This study examined the electrical axis of the fetal heart between 18 and 24 weeks gestation in a cohort study. This was a normative study in fetuses shown to have a normal fetal echo at a mean of 20 weeks gestation. Utilizing a non-invasive method with NEMO technology resulted in 6 dipolar electrodes in a standardized array on the maternal abdomen. Fetal echo orientation of the body position allowed the generation of a fetal vectorcardiogram from which a mean frontal fetal cardiac electric axis was generated equaling 123 degrees with 90% PI of 26 – 271 degrees. They concluded that this is consistent with the expected rightward electrical axis of the normal fetus and that they have generated reference values. While there is indeed a great need to perfect non-invasive fetal ecg’s for research and clinical purposes, and the authors are making a great contribution to science by developing this technology, unfortunately, the very wide confidence intervals of the mean value, even with such a large cohort, implies that this particular parameter is not clinically useful as is, and this is acknowledged by the authors. However, they are encouraged to continue to explore the utility of this technique. In fact, they have included a copy of their large long-term pilot study of fetal non-invasive electrocardiograms, of which this project is a small part.

It was unexpected that the electrical axis turned out to be so variable in this study. I posit that the greatest source of error is in their estimation of the fetal chest position. The mean frontal electric cardiac axis is very dependent on the orientation of the heart in relation to the 3-dimensional thorax. Postnatally, our electrode arrays are oriented to the usual levocardia, can be corrected for dextrocardia, and are known to vary with altered positional anatomy of the heart (eg, mesocardia) and the thorax (eg, scoliosis). The study’s method of having trainees periodically assess the fetal lie during the ecg, while the fetus may be moving, is the probable source of error. It is neither accurate nor precise. I would humbly suggest that they try this again, but next time carefully correct the vectorcardiogram for a fixed, anatomically and easily identified landmark, such as diaphragm, or even spine. Shortening the length of study might also reduce error due to fetal movement.

A few questions and comments:

Was this a convenience sample, or all consecutive normal fetal echos?

The technical success rate was remarkable in that only 5 of 281 eligible fetuses could not be analyzed.

The Title and Abstract are good.

The Introduction is long, but clear and informative.

The Methods are adequate, and while they mostly refer to their previous publications, these and the Research Protocol are all readily available.

The Results are straightforward.

The Discussion is a bit lengthy, but complete, including strengths, weaknesses, and future directions.

The text is clear and well written, but still needs careful proofreading.

This group has many prior publications in this field, which is supportive.

The References are comprehensive and relevant.

The Figures and Tables are good.

Overall, the science appears to be valid, technically believable and thus reproducible. The results are unique. Again, I encourage the authors to improve their correction for fetal thorax orientation.

Specific Comments:

The copy of the manuscript provided to me somehow included all of the preceding editorial markups and corrections. In addition, the text version I accessed included prior assessments by 3 previous reviewers. However, I generated my review before reading the other opinions.

Reference 30 is a book and appears incomplete without the name and location of the publisher.

The word “data” is always plural.

The sentence on lines 163-165 is unclear to me.

Answer: Thank you for your thorough reading of the paper and your feedback.

We want to apologize for having submitted a version with editorial markups and assessments from previous reviewers.

We admit that the determination of the fetal orientation is a limitation of this study. The fetal orientation was determined following a protocol in which the spine was taken as an identifiable landmark. The ultrasound probe was held only in a horizontal and vertical position for it to be reproducible and annotations were made about the position of the probe. A few factors can contribute to inaccuracies in the correction for the fetal orientation: inaccuracies in the correction for fetal movement and inaccurate assessment of the fetal orientation and the time at which this assessment was made.

To minimize these inaccuracies, we have used the ECG data that was recorded around the time that the fetal orientation assessment was done. Possible cumulative errors in the correction for movement will therefore be limited. As an extra check, we did determine the fetal orientation multiple times per recording. Within a single patient, the correct ECGs/VCGs are compared to verify their consistency, which can act as an indirect evaluation of the accuracy of our orientation correction. We have modified the manuscript to include this extra check.

With regard to the question regarding the samples that we used in our study. All measurements were taken of healthy fetuses after the 20 week anomaly scan, so these were all consecutive normal echos

Answer: Reference 30 has been completed. 

Answer: Lines 163-165 have been adjusted for more clarification. 

Reviewer #2: Thank you for the opportunity to read this interesting paper about fetal ECG´s performed in normal pregnancies. The authors conclude, that the range of a normal fetal heart axis is wide and the fetal heart axis alone is not suitable for screening of CHD due to the wide prediction interval. I agree to the comments of the three previous reviewers. The comments of the authors and their changes within the text improved the paper. In conclusion, the authors showed that despite an optimal setting (study setting, optimal time window) the fetal heart axis is wide [and its use not applicable for screening of CHD].

Although they did not include fetuses with CHD als reference group, it many be reasonable to draw the above mentioned conclusions.

This should be published.

comment to the revision: lines 292 to 299: The statement that the heart axis has been shown to rotate in a fetus with aortic stenosis or critical pulmonary stenosis is an assumption that has not yet been proven. A fetal echo has a significantly greater significance here. I recommend deleting this section.

Answer: Thank you for your positive comments regarding the manuscript. We agree with the suggestion on lines 292 to 299 and have deleted these. 

Reviewer #3: Lempersz et al performed a prospective study to evaluate the normal cardiac axis in fetuses.

Abstract:

This study does not evaluate electrical heart axis in CHD, it only evaluates normal fetuses. Please do not put in the discussion anything regarding the electrical axis in CHD screening.

Answer: Thank you for the time taken to read the manuscript and to provide valuable feedback. 

We have adjusted the discussion section of the abstract by deleting the sentence about CHD screening. 

Introduction:

Please cite: NI-fECG enables the production of a 12-lead electrocardiogram by means of a standardized method.

Answer: We have added a reference (no 14) for this.

Methods:

Study population, how do doctors evaluate cardiac health? Do they do echocardiograms? 

I find this paragraph from 124-130 confusing. Either describe the protocol in full or leave it out.

Answer: Before women were included in the study, they all underwent the fetal anomaly scan in which, amongst others, fetal cardiac health was assessed. Women could enter the study if the fetal anomaly scan showed a healthy fetus. The Paragraph (lines 124-130) has been changed for clarification. Postnatal check-ups consist of physical examination and auscultation of the heart.

Results/conclusions:

I think the major limitation of this study is the utility of this information. I think using this information to then compare to patients with CHD should be the next step and would make this a much more robust and useful study. Otherwise, the study should be reframed as a "proof of concept" study since we know that neonates have a rightward axis. I don't think any information regarding CHD should be put in the paper, because it misleads the reader into thinking that there will be some information on the fetal electrical axis in CHD.

Answer: Thank you for the useful feedback. We agree with this comment, that was also raised by one of the other reviewers. We have omitted the paragraph regarding CHD.

Further comments:

Previous edits are seen in the PDF, which made it challenging to review.

Answer: We apologize for having submitted the wrong version.

---

## [Decision Letter · Decision Letter 1]

11 Jun 2021

PONE-D-21-02858R1

The electrical heart axis of the fetus between 18 and 24 weeks of gestation: a cohort study.

PLOS ONE

Dear Dr. Lempersz,

Thank you for submitting your manuscript to PLOS ONE. After careful consideration, we feel that it has merit but does not fully meet PLOS ONE’s publication criteria as it currently stands. Therefore, we invite you to submit a revised version of the manuscript that addresses the points raised during the review process.

Please address the issues raised by the statistical reviewer. Thank you.

We look forward to receiving your revised manuscript.

Kind regards,

Gabor Erdoes, M.D., Ph.D.

Academic Editor

PLOS ONE

Journal Requirements:

Reviewers' comments:

Reviewer's Responses to Questions

**Comments to the Author**

1. If the authors have adequately addressed your comments raised in a previous round of review and you feel that this manuscript is now acceptable for publication, you may indicate that here to bypass the “Comments to the Author” section, enter your conflict of interest statement in the “Confidential to Editor” section, and submit your "Accept" recommendation.

Reviewer #1: All comments have been addressed

Reviewer #2: All comments have been addressed

Reviewer #4: (No Response)

2. Is the manuscript technically sound, and do the data support the conclusions?

Reviewer #1: Yes

Reviewer #2: Yes

Reviewer #4: Partly

3. Has the statistical analysis been performed appropriately and rigorously? 

Reviewer #1: Yes

Reviewer #2: Yes

Reviewer #4: No

4. Have the authors made all data underlying the findings in their manuscript fully available?

Reviewer #1: Yes

Reviewer #2: Yes

Reviewer #4: No

5. Is the manuscript presented in an intelligible fashion and written in standard English?

Reviewer #1: Yes

Reviewer #2: Yes

Reviewer #4: Yes

6. Review Comments to the Author

Reviewer #1: (No Response)

Reviewer #2: No further comments

The authors responded to all questions raised.

The manuscript has been reviewed an corrected.

Reviewer #4: PONE-D-21-02858R1: statistical review

SUMMARY. In this study, the electrical heart axis is proposed as a predictor of healthy fetuses around 20 weeks of gestation. The statistical analysis is based on a sample of frontal angles. Although the methods seem correct and the results are sound, some statistical computations are nonstandard because they belong to circular statistics. As such, these methods deserve a larger description than that one given by this version of the paper: see the specific issues below.

SPECIFIC ISSUES

1. The average frontal axis is a circular mean. Was it computed by the traditional ratio of trigonometric moments? Please clarify.

2. It seems that the confidence interval of the average frontal axis has been computed by assuming that the data are sampled from a von Mises distribution. Please provide the value of the concentration parameter of the von Mises distribution that has been used for computing the confidence interval.

3. Please clarify whether the von Mises parameters have been estimated by maximum likelihood methods. In this case, please clarify if the bias of the maximum likelihood estimate of the concentration parameter has been corrected or not.

3. The von Mises density is just one of the possible distributions that one can choose for circular data modelling. Without asking for a formal test of goodness of fit, could the authors at least overlap the estimated density on the empirical rose diagram of the data? This will show whether the von Mises provides a reasonable fit in this case study.

7. PLOS authors have the option to publish the peer review history of their article (what does this mean?). If published, this will include your full peer review and any attached files.

Reviewer #1: No

Reviewer #2: **Yes: **Ulrike Herberg

Reviewer #4: No

---

## [Author Response · Author response to Decision Letter 1]

23 Jul 2021

AUTHOR’s RESPONSE:

Reviewer #4: PONE-D-21-02858R1: statistical review

SUMMARY. In this study, the electrical heart axis is proposed as a predictor of healthy fetuses around 20 weeks of gestation. The statistical analysis is based on a sample of frontal angles. Although the methods seem correct and the results are sound, some statistical computations are nonstandard because they belong to circular statistics. As such, these methods deserve a larger description than that one given by this version of the paper: see the specific issues below.

SPECIFIC ISSUES

1. The average frontal axis is a circular mean. Was it computed by the traditional ratio of trigonometric moments? Please clarify.

 Yes, we took the maximum likelihood estimate (mle) of the mean direction parameter μ of a von Mises distribution as the average frontal axis which is the same as the first sample trigonometric moment: θ ®="atan2"(S ®/C ®), where C ®=1/n ∑_(j=1)^n▒cos⁡〖θ_j 〗 and S ®=1/n ∑_(j=1)^n▒sin⁡〖θ_j 〗⁡ . The praragraph has been adjusted in the manuscript (lines 176-179).

2. It seems that the confidence interval of the average frontal axis has been computed by assuming that the data are sampled from a von Mises distribution. Please provide the value of the concentration parameter of the von Mises distribution that has been used for computing the confidence interval.

3. Please clarify whether the von Mises parameters have been estimated by maximum likelihood methods. In this case, please clarify if the bias of the maximum likelihood estimate of the concentration parameter has been corrected or not.

 We would like to answer the reviewer’s comments 2-3 together since they are related to each other. First, we would like to mention that we computed the prediction interval for the frontal axis rather than the confidence interval. Nevertheless, reviewer 4’s comments regarding the bias of the concentration parameter κ are still valid. Indeed, we used the mle of the κ without bias corrections for the calculation of the prediction interval. Albeit the fact that the correction method proposed by Best and Fisher (1981) is still biased for small values of κ when the sample size is not large, we were able to verify that prediction interval based on the uncorrected κ ^ has close-to-nominal coverage probability via simulations (see Table 1 below). In this simulation study, for each simulation run, we drew 281 independent random samples from a von Mises distribution with parameter μ=4.142 and κ=0.526 (these values are also parameter estimates of the real data which the reviewer requested) and estimated the parameters using maximum likelihood. Based on the estimated parameters, we computed the corresponding quantiles of the von Mises distribution as our prediction interval. Thereafter, an independent sample was again drawn from the same distribution as a “future” observation. Across 10000 simulation runs, we calculated how often the prediction intervals contained the “future” observations.

Prediction interval Coverage probabilities

99% 98.95%

98% 98.02%

97% 96.72%

96% 96.06%

95% 95.11%

94% 93.98%

93% 93.05%

92% 91.46%

91% 90.37%

90% 90.33%

4. The von Mises density is just one of the possible distributions that one can choose for circular data modelling. Without asking for a formal test of goodness of fit, could the authors at least overlap the estimated density on the empirical rose diagram of the data? This will show whether the von Mises provides a reasonable fit in this case study.

 We agree with the reviewer that the von Mises distribution is just one of many possible distributions. We hereby provide the plot of the estimated density superimposed on the circular diagram. (diagram can be found in the 'response to reviewer' file)

---

## [Decision Letter · Decision Letter 2]

2 Aug 2021

The electrical heart axis of the fetus between 18 and 24 weeks of gestation: a cohort study.

PONE-D-21-02858R2

Dear Dr. Lempersz,

We’re pleased to inform you that your manuscript has been judged scientifically suitable for publication and will be formally accepted for publication once it meets all outstanding technical requirements.

Kind regards,

Gabor Erdoes, M.D., Ph.D.

Academic Editor

PLOS ONE

Reviewers' comments:

Reviewer's Responses to Questions

**Comments to the Author**

1. If the authors have adequately addressed your comments raised in a previous round of review and you feel that this manuscript is now acceptable for publication, you may indicate that here to bypass the “Comments to the Author” section, enter your conflict of interest statement in the “Confidential to Editor” section, and submit your "Accept" recommendation.

Reviewer #4: All comments have been addressed

2. Is the manuscript technically sound, and do the data support the conclusions?

Reviewer #4: (No Response)

3. Has the statistical analysis been performed appropriately and rigorously? 

Reviewer #4: (No Response)

4. Have the authors made all data underlying the findings in their manuscript fully available?

Reviewer #4: (No Response)

5. Is the manuscript presented in an intelligible fashion and written in standard English?

Reviewer #4: (No Response)

6. Review Comments to the Author

Reviewer #4: (No Response)

7. PLOS authors have the option to publish the peer review history of their article (what does this mean?). If published, this will include your full peer review and any attached files.

Reviewer #4: No

---

## [Editor Report · Acceptance letter]

7 Dec 2021

PONE-D-21-02858R2 

The electrical heart axis of the fetus between 18 and 24 weeks of gestation: a cohort study. 

Dear Dr. Lempersz:

I'm pleased to inform you that your manuscript has been deemed suitable for publication in PLOS ONE. Congratulations! Your manuscript is now with our production department. 

Kind regards, 

on behalf of

Prof. Dr. Dr. Gabor Erdoes 

Academic Editor

PLOS ONE